# Saddlepoint Method for Pricing European Options under Markov-Switching Heston's Stochastic Volatility Model

**Mengzhe Zhang** [1] and **Leunglung Chan** [2,*]

1   Psychometrics and Analytics Branch, NSW Education Standards Authority, Sydney, NSW 2001, Australia
2   School of Mathematics and Statistics, University of New South Wales, Sydney, NSW 2052, Australia
*   Correspondence: leung.chan@unsw.edu.au

**Abstract:** This paper evaluates the prices of European-style options when dynamics of the underlying asset is assumed to follow a Markov-switching Heston's stochastic volatility model. Under this framework, the expected return and the long-term mean of the variance of the underlying asset rely on states of the economy modeled by a continuous-time Markov chain. There is evidence that the Markov-switching Heston's stochastic volatility model performs well in capturing major events affecting price dynamics. However, due to the nature of the model, analytic solutions for the prices of options or other financial derivatives do not exist. By means of the saddlepoint method, an analytic approximation for European-style option price is presented. The saddlepoint method gives an effective approximation to option prices under the Markov-switching Heston's stochastic volatility model.

**Keywords:** European-style options; Markov-switching Heston's stochastic volatility model; saddlepoint method; Markov chain

## 1. Introduction

It is well documented that the stochastic volatility (SV) models take the volatility smile effect into account in the real markets. Needless to say, the SV models are more realistic than the standard Black–Scholes model. Nevertheless, Vo (2009) conducted an empirical study on a Markov-switching stochastic volatility model and showed that the behaviour of crude oil price is well explained under the Markov-switching SV model. A significant regime-switching effect on the oil markets is well documented and their numerical results reveal that the forecasting power of the SV model with Markov switching outperforms its non-switching counterpart. The empirical study also finds that the SV model with Markov switching is able to capture large shocks of the oil markets well. These advantages of empirical findings give us a motivation to pricing options under SV models with Markov switching.

This paper shows how to price a European-style option in Markov-switching Heston's stochastic volatility model. Under the Markov-switching Heston's stochastic volatility model, the key parameters of the model follow Markov chains which switch from a state to another state as time evolves. The Markov-switching (regime-switching) behaviour could reflect the dynamic preferences and time-varying beliefs of traders or the changing levels of economy activities. In general, the Markov chain is unobservable and the chain is modeled by a stochastic differential equation (SDE). Zhu et al. (2012) derive an explicit European option pricing formulas in a two-state regime by solving a couple of partial differential equations (PDEs) using the Fourier transform method. Chan and Zhu (2021b) obtain a closed-form formulas of Lookback options under a Markov-switching Wiener process. Chan and Zhu (2015) derive a closed-form formula of American convertible bonds in a Black–Scholes–Merton's model with regime switching. Chan and Zhu (2021a) obtain an analytic approach for pricing American options with Markov switching. Elliott et al.

(2013) price options under the CEV model with Markov switching. Chan and Zhu (2015) consider a homotopy analysis method to price barrier options under the Markov-switching Wiener process. Elliott et al. (2014) use a quadratic approximation approach to price barrier options under the Markov switching model. Bollen (1998) calculates a European option price under the Markov switching model via a binomial tree. Hardy (2001) proposes a recursive algorithm to price European options under the Markov switching model. Duan et al. (2002) use a lattice-approximation method to price European-style and American-style options under the Markov switching model. Boyle and Draviam (2007) use a finite difference scheme to price exotic options under the Markov switching model. Li et al. (2012) derive the bounds of exotic option price using semidefinite programming (SDP) under the Markov switching model. Lu and Putri (2020) evaluate Markov-switching American option via Laplace transform. Egami and Kevkhishvili (2020) simplify an optimal stopping problem for a two-state Markov-switching model to a pair of no-switching optimal stopping problems. Elliott and Lian (2013) investigate the pricing of variance swaps under a Markov-switching stochastic volatility model.

The saddlepoint method has been widely used in approximating the probability distribution of the partial sum of independent random variables since it was proposed by Daniels in 1954. In this paper, instead of approximating distributions, we use the Lugannani and Rice (LR) formula (Daniels 1987) to approximate tail probabilities of logarithm of the underlying asset. In order to obtain the price dynamics under a martingale measure, we adopt an Esscher transform (Elliott et al. 2005). Once we obtain the closed-form cumulant generating functions (CGFs) of $\ln(S_t)$ under the martingale measure $Q$ and the physical probability measure $P$, respectively, then the saddlepoint equation can be solved via any symbolic computing package, such as Maple. The remaining procedure for pricing options is just simple algebraic manipulations. Glasserman and Kim (2009) use saddlepoint methods to calculate a European call option in a jump-diffusion framework. Zhang and Chan (2016) obtain saddlepoint approximation for European call options in a Markov-switching model and our current work extends their work to a Markov-switching SV model.

The paper is organized as follows. Section 2 reviews Markov-switching Heston's stochastic volatility model. Section 3 discusses the saddlepoint method and LR formula. Section 4 derives the cumulant generating functions of the model under different probability measures. Sections 5 and 6 refer to numerical results and conclusion, respectively.

## 2. Markov-Switching Heston's Stochastic Volatility Model

After adopting an Esscher transform (Elliott et al. 2005), the dynamics of underlying asset and the volatility in the Markov-switching Heston's stochastic volatility model under a martingale measure, $\mathcal{P}$, are assumed to follow the following SDEs:

$$
\begin{aligned}
dS_t &= r_t S_t dt + \sqrt{\sigma_t} S_t dW_t^S, \quad &(1)\\
d\sigma_t &= \kappa(\theta_t^\star - \sigma_t)dt + \sigma_v \sqrt{\sigma_t} dW_t^\sigma. \quad &(2)
\end{aligned}
$$

Here $r_t$ is the short rate, $\theta_t$ is the long-term mean of the variance, $\kappa$ is a mean-reverting speed parameter of the variance, $\sigma_v$ is the volatility of volatility, $\mu_t$ is the mean rate of return and $\theta_t^\star := \theta_t - \rho \sigma_v(\mu_t - r_t)$. $(dW_t^S, dW_t^\sigma)$ is a two-dimensional Brownian motion with $\langle dW_t^S, dW_t^\sigma \rangle = \rho t$. We define $Y_t = \ln(S_t)$ and apply Ito's formula to $Y_t$ to get a new process $(Y, \sigma)$ with

$$
\begin{aligned}
dY_t &= (r_t - \frac{1}{2}\sigma_t)dt + \sqrt{\sigma_t} S_t dW_t^S, \quad &(3)\\
d\sigma_t &= \kappa(\theta_t^\star - \sigma_t)dt + \sigma_v \sqrt{\sigma_t} dW_t^\sigma. \quad &(4)
\end{aligned}
$$

From Heston (1993), the value of a European call option is in form of

$$
\begin{aligned}
C(t,S) &= E^Q \left[ \exp(-r(T-t))(S_T - K)^+ \right]\\
&= S_0 \mathcal{Q}(Y_T > \ln(K)) - K \exp(-r(T-t))\mathcal{P}(Y_T > \ln(K)). \quad (5)
\end{aligned}
$$

Here $\mathcal{Q}$ is defined by the measure change $\frac{d\mathcal{Q}}{d\mathcal{P}} = e^{-r(T-t)}e^{Y_T-Y_0}$, which uses $S_T$ as a numeraire asset. The dynamics of $(Y, \sigma)$ can be written as

$$dY_t = (r_t + \frac{1}{2}\sigma_t)dt + \sqrt{\sigma_t}S_t dW_t^{S,Q}, \tag{6}$$

$$d\sigma_t = (\kappa\theta_t^\star - (\kappa - \rho\sigma_v)\sigma_t)dt + \sigma_v\sqrt{\sigma_t}dW_t^{\sigma,Q}. \tag{7}$$

Here $W_t^{S,Q}$ and $W_t^{\sigma,Q}$ are standard Brownian motions under $\mathcal{Q}$ with a correlation parameter $\rho$. To make the parameters depend on the chains, we introduced the notation in the following parameters $\mu_t$, $\theta_t$ and $r_t$ where

$$\mu_t := \langle\mu, \mathbf{X}_t\rangle,$$
$$r_t := \langle r, \mathbf{X}_t\rangle,$$
$$\theta_t := \langle\theta, \mathbf{X}_t\rangle.$$

Here $\langle.\rangle$ denotes an inner product. $\mathbf{X}_t$ is a continuous-time Markov chain. We assume that the Markov chain $\mathbf{X}$ is irreducible. Without loss of generality, we can identify the state space of the chain $\mathbf{X}$ with the finite set of unit vectors $\mathcal{E} := \{\mathbf{e}_1, \mathbf{e}_2, \cdots, \mathbf{e}_N\}$, where $\mathbf{e}_i := (0, \cdots, 1, \cdots, 0)' \in \Re^N$. From Elliott et al. (1994), the semi-martingale representation for the chain is given by $\mathbf{X}$:

$$\mathbf{X}_t = \mathbf{X}_0 + \int_0^t \mathbf{A}'\mathbf{X}_t du + \mathbf{M}_t. \tag{8}$$

Here $\mathbf{M}_t$ is a martingale and $\mathbf{A}'$ is the transpose of $\mathbf{A}$. $\mathbf{A}' := [\lambda_{ij}]_{i,j=1,2,\cdots,N}$ denotes the intensity matrix of the chain $\mathbf{X}$, $\lambda_{ij}$ is the constant rate of transition of the chain $\mathbf{X}$ from state $\mathbf{e}_i$ to state $\mathbf{e}_j$. For more details about the Markov-switching model, see Guo (2001) and Buffington and Elliott (2002).

## 3. Saddlepoint Methods

The saddlepoint approximation was proposed by Daniels in 1954 and is used to approximate the probability density function of the sum $\bar{Y} = \sum_1^n Y_k/n$, where $Y_k's$ are identically independent distributed random variables. Assume the cumulant generating function $G(z)$ of $\bar{Y}$ is known, the probability density $f_n(\bar{y})$ of $\bar{Y}$ is in the form of

$$f_n(\bar{y}) = \frac{n}{2\pi i}\int_{\tau-i\infty}^{\tau+i\infty} \exp\left(n\big(G(z) - z\bar{y}\big)\right)dz, \tag{9}$$

for any $\tau \in \{y \in \Re : |G(\bar{y})| < \infty\}$. Through the steepest descent method, Daniels (1954) gave an asymptotics for the integral (9). The approximation $\hat{f}_n(\bar{y})$ could be computed by selecting the path to pass through the saddle point $\bar{z}$ such that $G'(\bar{z}) - \bar{y} = 0$. The modulus of the integrand attains a maximum at $\bar{z}$ as the function $G(z) - z\bar{y}$ has a minimum at $\bar{z}$. While the integrand does not make a contribution apart from a neighbourhood of the saddlepoint. Consequently, higher order terms in the expansion can be ignored without losing much precision.

Daniels (1987) obtained a probability approximation formula based on the Lugannani–Rice (LR) formula. Compared to those of the Edgeworth expansion, the error of the LR formula is almost uniformly distributed over the whole range of the expectation $\bar{y}$. The LR formula is given by

$$P(\bar{Y} > \bar{y}) = 1 - \Phi(\sqrt{n}\hat{w}) + \phi(\sqrt{n}\hat{w})\left\{\frac{b_0}{n^{1/2}} + \frac{b_1}{n^{3/2}} + o(n^{-3/2})\right\}, \tag{10}$$

where $b_0 = 1/\hat{u} - 1/\hat{w}$, $b_1 = (\lambda_4/8 - 5\lambda_3^2/24)/\hat{u} - \lambda_3/(2\hat{u}^2) - 1/\hat{u}^3 + 1/\hat{w}^3$, $\hat{w} = sgn(\bar{z})\sqrt{2(\bar{z}y - G(\bar{z}))}$, $\hat{u} = \bar{z}\sqrt{G''(\bar{z})}$, $\lambda_3 = G^{(3)}(\bar{z})/G''(\bar{z})^{3/2}$ and $\lambda_4 = G^{(4)}(\bar{z})/G''(\bar{z})^{4/2}$.

Here $\Phi$, $\phi$ are the cumulative distribution function and the probability density of the standard normal distribution, respectively.

## 4. Pricing European Options in Markov-Switching Heston's Model

If the underlying asset follows dynamics (1) and (2), the price of a European call option at time zero is in the form of

$$
\begin{aligned}
C(t,S) &= E^Q\left[\exp(-r(T-t))(S_T-K)^+\right]\\
&= S_0\mathcal{Q}(Y_T>\ln(K))-K\exp(-r(T-t))\mathcal{P}(Y_T>\ln(K))\,. \quad (11)
\end{aligned}
$$

The CGF of $Y_T$ under $\mathcal{P}$ is denoted by $e^{G(z,Y,t,\sigma)}=E[e^{zY_T}]$. Denote $\mathcal{F}^S_{jt}$, $\mathcal{F}^\sigma_{jt}$ and $\mathcal{F}^X_t$ by the natural filtrations generated by the Brownian motions $W^S_t$, $W^\sigma_t$ and the Markov chain $X_t$, up to time $t$, respectively, we have

$$
\begin{aligned}
\mathcal{F}^S_{1t} &:= \hat\sigma\{W^S_u|u\le t\},\\
\mathcal{F}^\sigma_{1t} &:= \hat\sigma\{W^\sigma_u|u\le t\},\\
\mathcal{F}^X_t &:= \hat\sigma\{X_u|u\le t\}
\end{aligned}
$$

and

$$
\begin{aligned}
\mathcal{F}^S_{2t} &:= \hat\sigma\{W^{S,Q}_u|u\le t\},\\
\mathcal{F}^\sigma_{2t} &:= \hat\sigma\{W^{\sigma,Q}_u|u\le t\}.
\end{aligned}
$$

Here $\hat\sigma$ is the smallest $\sigma$-field. To obtain the CGF we calculate the expectation by enlarging the filtration, where we assume $\mathcal{F}^X_T$ is known. In such a case, all parameters depending on the Markov chain $X$ would degenerate to deterministic functions of time. For instance, given $F^X_T$, the CGF is given by the following lemma.

**Lemma 1.** *If the underlying asset follows the dynamics (1) and (2), and $\mathcal{F}^X_T$ is given, then the CGF of the stochastic variable $Y_T = \ln(S_T)$ is given by*

$$
\begin{aligned}
f_j(z,Y,t,\sigma)|\mathcal{F}^X_T) &= E^Q\left[e^{zY_T}|\mathcal{F}^S_{jt}\vee\mathcal{F}^\sigma_{jt}\vee\mathcal{F}^X_T\right]\\
&= e^{F(z,t,T)+H(z,t,T)\sigma+zY}, \quad (12)
\end{aligned}
$$

*where $F(z,t,T)$ and $H(z,t,T)$ are given by*

$$
\begin{aligned}
F(z,t,T) &= \int_t^T\langle rz+\kappa\theta^\star H(z,t,T),\mathbf{X_s}\rangle ds,\\
H(z,t,T) &= \frac{b_j-\rho\sigma_v z+d}{\sigma_v^2}\left[\frac{1-e^{d(T-t)}}{1-ge^{d(T-t)}}\right],\\
g &= \frac{b_j-\rho\sigma_v z+d}{b_j-\rho\sigma_v z-d},\\
d &= \sqrt{(\rho\sigma_v z-b_j)^2-\sigma_v^2(2u_{jz}+z^2)},\\
b_1 &= \kappa, b_2=\kappa-\rho\sigma_v,\\
u_1 &= -\frac{1}{2}, u_2=\frac{1}{2}.
\end{aligned}
$$

**Proof.** When the price dynamics follow (1) and (2), Heston (1993) demonstrates that the value of any asset $U(S,\sigma,t)$ must satisfy the partial differential equation (PDE)

$$\frac{1}{2}\sigma S^2 \frac{\partial^2 U}{\partial S^2} + \rho\sigma\sigma_v S \frac{\partial^2 U}{\partial S \partial \sigma} + \frac{1}{2}\sigma_v^2\sigma\frac{\partial^2 f}{\partial \sigma^2} + rS\frac{\partial U}{\partial S} + \kappa(\theta^\star(t) - \sigma)\frac{\partial U}{\partial \sigma} \quad - \quad rU + \frac{\partial U}{\partial t} = 0. \quad (13)$$

Thus we guess the solution of the model has the form

$$C(S,\sigma,t) = SP_2 - Ke^{-rT}P_1. \quad (14)$$

Both of $P_1$ and $P_2$ must satisfy the original PDE (13). Substituting the proposed solution (14) into the original PDE (13) shows that $P_1$ and $P_2$ must satisfy the PDEs

$$\frac{1}{2}\sigma\frac{\partial^2 P_j}{\partial Y^2} + \rho\sigma\sigma_v S\frac{\partial^2 P_j}{\partial Y \partial \sigma} + \frac{1}{2}\sigma_v^2\sigma\frac{\partial^2 P_j}{\partial \sigma^2} + (r + u_j\sigma)\frac{\partial U}{\partial Y} + (\kappa\theta_t^\star - b_j\sigma)\frac{\partial U}{\partial \sigma} + \frac{\partial U}{\partial t} = 0, \quad (15)$$

for $j = 1, 2$ and where $b_1 = \kappa$, $b_2 = \kappa - \rho\sigma_v, u_1 = -\frac{1}{2}$, $u_2 = \frac{1}{2}$. $P_j$ could be treated as the conditional probability that the option expires in-the-money,

$$P_j(Y,\sigma,T,\ln[K]) = \Pr[Y_T \geq \ln[K]|Y_t,\sigma_t,X_t]. \quad (16)$$

We can solve PDE (15) by first guessing that the affine form solution might be

$$f_j(z,Y,t,\sigma)|\mathcal{F}_T^X) = e^{F(z,t,T)+H(z,t,T)\sigma_t+zY_t}. \quad (17)$$

The functions $F(z,t,T)$ and $H(z,t,T)$ can be found by solving two Riccati ODEs:

$$-\frac{\partial F}{\partial t} = r_t z + \kappa\theta_t^\star F \quad (18)$$

$$-\frac{\partial H}{\partial t} = \frac{1}{2}z^2 + u_j z + (\rho\sigma_v z - b_j)H + \frac{1}{2}\sigma_v^2 H^2 \quad (19)$$

with initial conditions $F(z,t,T) = 0$ and $H(z,T,T) = 0$. The solutions are

$$F(z,t,T) = \int_t^T \langle rz + \kappa\theta^\star H(z,t,T), \mathbf{X_s}\rangle ds,$$

$$H(z,t,T) = \frac{b_j - \rho\sigma_v z + d}{\sigma_v^2}\Big[\frac{1 - e^{d(T-t)}}{1 - ge^{d(T-t)}}\Big],$$

$$g = \frac{b_j - \rho\sigma_v z + d}{b_j - \rho\sigma_v z - d},$$

$$d = \sqrt{(\rho\sigma_v z - b_j)^2 - \sigma_v^2(2u_{jz} + z^2)},$$

$$b_1 = \kappa, b_2 = \kappa - \rho\sigma_v,$$

$$u_1 = -\frac{1}{2}, u_2 = \frac{1}{2}.$$

□

**Lemma 2.** *If the underlying asset follows the dynamics (1) and (2), then the MGF of the stochastic variable $y(t,T) = \int_t^T \sigma(s)ds$ is given by*

$$f_j(z,Y,t,\sigma) = \langle\Phi(z,t,T)X_t, \mathbf{I}_d\rangle \times e^{H(z,t,T)\sigma_t+zY_t}, \quad (20)$$

*where*

$$
\begin{aligned}
\Phi(z,t,T) &= \exp(\mathbf{A}'(T-t) + \text{diag}[rz(T-t) \\
&+ \frac{\kappa\theta^\star}{\sigma_v^2}((b_j - \rho\sigma_v z - d)(T-t) - 2\ln[\frac{1 - e^{d(T-t)}}{1 - ge^{d(T-t)}}])]), \\
H(z,t,T) &= \frac{bj - \rho\sigma_v z + d}{\sigma_v^2}[\frac{1 - e^{d(T-t)}}{1 - ge^{d(T-t)}}], \\
g &= \frac{b_j - \rho\sigma_v z + d}{b_j - \rho\sigma_v z - d'} \\
d &= \sqrt{(\rho\sigma_v z - b_j)^2 - \sigma_v^2(2u_{jz} + z^2)}, \\
b_1 &= \kappa, b_2 = \kappa - \rho\sigma_v, \\
u_1 &= -\frac{1}{2}, u_2 = \frac{1}{2}.
\end{aligned}
$$

**Proof.** In order to derive the unconditional MGF, we need to calculate the expectation without conditioning, where $\theta^\star$ relies on the chain of $X$ process up to time $T$. Consequently, we write

$$
\begin{aligned}
f_j(z, Y, t, \sigma) &= E[e^{zY_T}|\mathcal{F}_t^S \vee \mathcal{F}_t^\sigma \vee \mathcal{F}_t^X] \\
&= E[E[e^{zY_T}|\mathcal{F}_t^S \vee \mathcal{F}_T^\sigma \vee \mathcal{F}_T^X]|\mathcal{F}_t^S \vee \mathcal{F}_t^\sigma \vee \mathcal{F}_t^X] \\
&= E[e^{F(z,t,T) + H(z,t,T)\sigma_t + zY_t}|\mathcal{F}_t^S \vee \mathcal{F}_t^\sigma \vee \mathcal{F}_t^X] \\
&= E[\int_t^T \langle rz + \kappa\theta^\star H(z,t,T), \mathbf{X_s}\rangle ds|\mathcal{F}_t^S \vee \mathcal{F}_t^\sigma \vee \mathcal{F}_t^X] \times e^{H(z,t,T)\sigma_t + zY_t}.
\end{aligned}
\tag{21}
$$

By the Proposition 3.2 in Elliott and Lian (2013), we have

$$
E[\int_t^T \langle rz + \kappa\theta^\star H(z,t,T), \mathbf{X_s}\rangle ds|\mathcal{F}_t^S \vee \mathcal{F}_t^\sigma \vee \mathcal{F}_t^X] = \langle \Phi(z,t,T)X_t, \mathrm{I}_d\rangle,
\tag{22}
$$

where $\Phi(z,t,T)$ is an $N$-by-$N$ $\Re$-valued matrix given by

$$
\begin{aligned}
\Phi(z,t,T) &= \exp\left(\int_t^T \mathbf{A}' + \text{diag}[rz + \kappa\theta^\star H(z,t,T)]ds\right) \\
&= \exp(\mathbf{A}'(T-t) + \text{diag}[rz(T-t) \\
&+ \frac{\kappa\theta^\star}{\sigma_v^2}((b_j - \rho\sigma_v z - d)(T-t) - 2\ln[\frac{1 - e^{d(T-t)}}{1 - ge^{d(T-t)}}])]\right)
\end{aligned}
\tag{23}
$$

$\mathrm{I}_d = (1,1,\dots 1) \in \Re^N$ and $\mathbf{A}'$ denotes the transpose of $\mathbf{A}$. So,

$$
f_j(z, Y, t, \sigma) = \langle \Phi(z,t,T)X_t, \mathrm{I}_d\rangle \times e^{H(z,t,T)\sigma_t + zY_t}.
\tag{24}
$$

□

At last $G_j(z, Y, t, \sigma) = \ln[f_j(z, Y, t, \sigma)] = \ln\langle\Phi(z,t,T)X_t, \mathrm{I}_d\rangle + H(z,t,T)\sigma_t + zY_t$. Using this CGF, we can apply the saddlepoint approximation formula presented in the previous section to calculate option price.

$$
C(S, \sigma, t) = SP_2 - Ke^{-rT}P_1.
\tag{25}
$$

## 5. Numerical Examples

In this section, we give some examples to price European-style call options under Markov-switching Heston's stochastic volatility model using the saddlepoint method. Without the loss of generality, we consider two-regime case: the first regime ($X_t = 1$) refers to the 'Booms' state of economy and the second regime ($X_t = 2$) refers to the 'Recessions'

state of economy. The parameters setting in our example are $S_0 = 100$, $\kappa = 2$, $\rho = -0.2$, $\sigma_v = 0.2$, $r = [0.03; 0.03]$, $\mu = [0.08; 0.04]$, and $\theta = [0.009; 0.004]$. The transition rate matrix of the Markov chain **A** is $[-1, 1; 1, -1]$. Assuming that the current state is $X_0 = 1$ and that the current volatility level is $V_0 = 0.04$. The saddlepoint can be found by solving the saddlepoint equation, $G'_j(z) - \ln[K] = 0$, through symbolic computing softwares like Mathematica or Maple. The results for a two-regime case are documented in Table 1. Here $MC_i$ denotes Monte-Carlo results based on current state $i$. We set sample paths of Monte-Carlo simulation to 20,000 and the computation is performed using Matlab R2010b programs as the benchmark. $SA1_i$ and $SA2_i$ refers to the ASP method based on the current state $i$ with first order and second order, respectively. The results show that '$SA2_i$' would definitely outperform '$SA1_i$' for all range of maturity. Table 2 demonstrates option prices with different strikes under Markov-switching Heston's stochastic volatility model as well as its non-switching Heston's counterpart. In a non-switching Heston's model, we assume all parameters' value equals the parameters' value in the first regime of the Markov-switching Heston's model. The numerical results are shown in Figure 1 and it clearly shows that our approximation is quite precise. We also notice that the option prices calculated from the Markov-switching Heston's model are cheaper than their non-switching counterparts over all range of strike prices given in the Table 2. This is because parameters' value in a regime-switching Heston's model is smaller. As $\theta_1 < \theta_2$ and $\mu_1 < \mu_2$, so $\frac{1}{T} \int \theta_1 dt > \frac{1}{T} \int \theta_t dt$ and $\frac{1}{T} \int \mu_1 dt > \frac{1}{T} \int \mu_t dt$. Figure 2 is the plot of the implied volatility. The maturity time is fixed at T = 1 and leaves other parameters unchanged. When $\rho$ is zero we get a volatility smile and the implied volatility starts to increase after the strike price is greater than the initial stock price. A negative $\rho$ has an obvious impact on the shape of the volatility curve: the volatility smile becomes a smirk. It seems that the changes in the initial regime only has an impact on the values of implied volatility, but not the shape of volatility curve.

**Table 1.** Call option prices (N = 2).

| | | | **First** | **Order** | **Sec** | **Order** |
|---|---|---|---|---|---|---|
| *T* (year) | $MC_1$ | $MC_2$ | $SA1_1$ | $SA1_2$ | $SA2_1$ | $SA2_2$ |
| 0.1 | 8.5315 | 8.2114 | 8.4556 | 8.0806 | 8.5043 | 8.1738 |
| 0.2 | 8.4257 | 7.8141 | 8.0043 | 7.3541 | 8.3253 | 7.7503 |
| 0.5 | 8.0846 | 7.1332 | 7.2920 | 6.2731 | 7.9300 | 7.0056 |
| 1 | 7.7494 | 6.7888 | 6.8765 | 5.8592 | 7.5898 | 6.6773 |

**Table 2.** Call option prices when T = 1, N = 2.

| | **with Regime-Switching** | **with Regime-Switching** | **without Regime-Switching** |
|---|---|---|---|
| K | $MC_1$ | $SA2_1$ | First State |
| 70 | 32.0532 | 32.1250 | 32.3157 |
| 80 | 22.8978 | 22.6910 | 23.3394 |
| 90 | 14.0194 | 14.0200 | 15.4829 |
| 100 | 7.1860 | 7.1398 | 9.3352 |
| 110 | 3.0066 | 2.7466 | 5.1183 |
| 120 | 1.0630 | 0.9057 | 2.5812 |
| 130 | 0.3573 | 0.2758 | 1.2193 |
| 140 | 0.1071 | 0.0829 | 0.1175 |

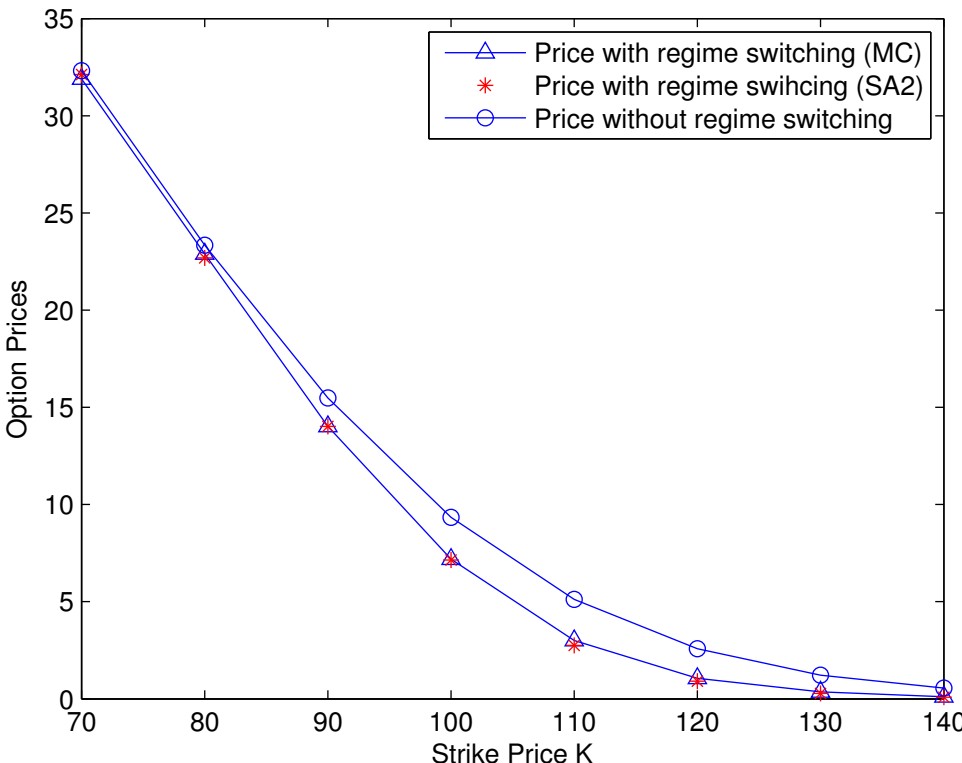

**Figure 1.** A picture of a Table 2.

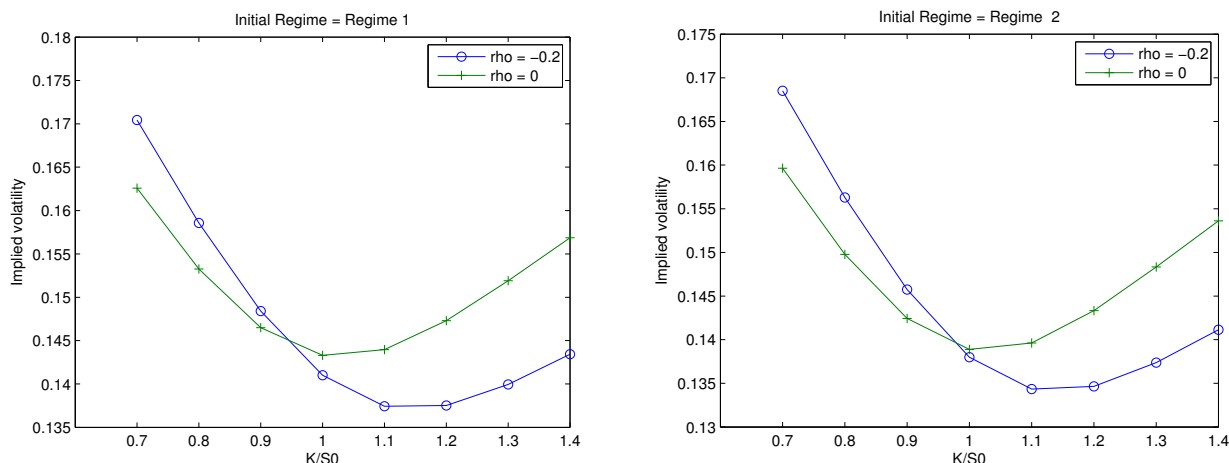

**Figure 2.** Volatility Smile (T = 1).

## 6. Conclusions

This paper studies a saddlepoint approximation approach for pricing options in a Markov-switching Heston's stochastic volatility model. Our approach requires to derive a closed-form expression of cumulant generating function, which needs to solve partial differential equations. The numerical results in Section 5 examine the accuracy of our method and a two-state case has been considered. The results show that saddlepoint method gives quite accuracy for the given range of maturities.

However, if the closed-form cumulant generating function does not exist, then one needs to rely on a numerical method to find a saddle point. In this situation, the procedure becomes slower.

In the future, it is worth considering multi-state Markov chain case and developing an algorithm to beat the curse of dimensionality. Furthermore, pricing American option under

Markov-switching Heston's stochastic volatility model and its calibration are interesting to study.

**Author Contributions:** Conceptualization, M.Z. and L.C. All authors have read and agreed to the published version of the manuscript.

**Funding:** This research received no external funding.

**Institutional Review Board Statement:** Not applicable.

**Informed Consent Statement:** Not applicables.

**Data Availability Statement:** Not applicable.

**Conflicts of Interest:** The authors declare no conflict of interest.

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
