# Peer review of "Saddlepoint Method for Pricing European Options under Markov-Switching Heston’s Stochastic Volatility Model"

_jrfm, doi:10.3390/jrfm15090396_

Round 1
Reviewer 1 Report
This paper extends the authors previous work in using saddlepoint methods to find (approximate) the price of an option under regime switching but under stochastic volatility under the Heston model. The presentation is clear and is easy to read and the results of the paper are a definite contribution. The only thing I would suggest to the authors is to cite some papers that take a different approach to finding approximations to option prices under regime switching as in
| A moment approach to bounding exotic options under regime switching
JY Li, MJ Kim, RH Kwon
Optimization 61 (10), 1253-1269
|
| 0 | 2012 | |
Author Response
Thank you very much for such a positive reviewer’s report with a suggestion. Your suggestion has made us add some sentences in the revised manuscript to make it much clearer to readers about a different approach to finding approximations to option prices under regime switching. For your convenience, we explain the changes we have made below, according to the point you have raised it, and have highlighted all the revised sentences in red in the manuscript.
Point 1: The only thing I would suggest to the authors is to cite some papers that take a different approach to finding approximations to option prices under regime switching as in
Li, J.Y., Kim, M.J. and Kwon, R.H., A moment approach to bounding exotic options under regime switching, Optimization, 2012, 61(10), pp.1253 - 1269.
Response 1: We have added some citations about different approach to approximate option prices under regime switching. We added some sentences below.
Elliott, Chan and Siu (2013) price options under the CEV model with regime switching. Chan and Zhu (2015) consider a homotopy analysis method to price barrier options under the regime switching model. Elliott, Siu and Chan (2014) use a quadratic approximation approach to price barrier options under the regime switching model. Bollen (1998) calculates a European option price under the regime switching model via a binomial tree. Hardy (2001) proposes a recursive algorithm to price European options under the regime switching model. Duan et al. (2002) use a lattice-approximation method to price European-style and American-style options under the regime switching model. Boyle and Draviam (2007) use a finite difference scheme to price exotic options under the regime switching model. Li et al. (2012) derive the bounds of exotic option price using semidefinite programming (SDP) under the regime switching model. Lu and Putri (2020) applied Laplace transform to regime-switching American option and obtained a semi-analytical solution. Egami and Kevkhishvili (2020) reduced an optimal stopping problem with an arbitrary value function in a two-regime environment to a pair of optimal stopping problems without regime switching.
Reviewer 3 Report
This paper studies the prices of European options when dynamics of the underlying asset is assumed to follow the Heston's model with regime switching. However, I have some major remarks as follows:
1) Citation (Vo, 2009) should not be stated in the abstract. The authors need to revise this statement.
2) Need to revise the abstract, by highlighting the main findings of this study.
3) It seems insufficient review on the past studies. Suggest the authors to include more papers (including the latest study) in this paper.
4) Suggest the authors to include a table for the “State of the art” to show the research gap clearly.
5) Some of the equations are not labelled.
6) Need to justify parameters used in the “Section 5: Numerical Results”. Is it based on the past studies or any reference? Different set of parameters may affect the results.
7) Need to include the limitation of the proposed method and future research work in the conclusion section.
8) The references are not up to date. Please include the latest 5 years of papers in the references. Please include and discuss the latest research work (year 2018-2021).
Author Response
Thank you very much for your suggestions and a couple of questions. Your suggestions and questions have made us add some sentences in the revised manuscript to make it much clearer to readers and help us make improvements of the current manuscript. For your convenience, we explain the changes we have made below, according to the points you have raised them, and have highlighted all the revised sentences in red in the manuscript.
- Citation (Vo, 2009) should not be stated in the abstract. The authors need to revise this statement.
Response 1: Thanks for the comment. We have revised this statement below.
There is evidence that the stochastic volatility model with regime switching performs well in capturing major events affecting price dynamics.
- Need to revise the abstract, by highlighting the main findings of this study.
Response 2: We have revised the abstract.
By means of the saddlepoint method, an analytic approximation for the price of European option is presented. The saddlepoint method gives an effective approximation to option prices under the regime switching model.
- It seems insufficient review on the past studies. Suggest the authors to include more papers (including the latest study) in this paper.
Response 3: We added some discussion about the past studies (please see red part in the manuscript).
Elliott, Chan and Siu (2013) price options under the CEV model with regime switching. Chan and Zhu (2015) consider a homotopy analysis method to price barrier options under the regime switching model. Elliott, Siu and Chan (2014) use a quadratic approximation approach to price barrier options under the regime switching model. Bollen (1998) calculates a European option price under the regime switching model via a binomial tree. Hardy (2001) proposes a recursive algorithm to price European options under the regime switching model. Duan et al. (2002) use a lattice-approximation method to price European-style and American-style options under the regime switching model. Boyle and Draviam (2007) use a finite difference scheme to price exotic options under the regime switching model. Li et al. (2012) derive the bounds of exotic option price using semidefinite programming (SDP) under the regime switching model. Lu and Putri (2020) applied Laplace transform to regime-switching American option and obtained a semi-analytical solution. Egami and Kevkhishvili (2020) reduced an optimal stopping problem with an arbitrary value function in a two-regime environment to a pair of optimal stopping problems without regime switching.
- Suggest the authors to include a table for the “State of the art” to show the research gap clearly.
Response 4: There is no one used the Saddlepoint method on Heston’s model with regime switching, it is not worth to include a table for the “State of the art” to show the research gap. To know the research gap, please read the review on the past studies.
- Some of the equations are not labelled.
Response 5: We have labelled the equations.
- Need to justify parameters used in the “Section 5: Numerical Results”. Is it based on the past studies or any reference? Different set of parameters may affect the results.
Response 6: Some parameters are based on the paper of Elliott and Lian (2013).
Elliott, R.J., and Lian, G., Pricing variance and
volatility swaps in a stochastic volatility model with regime switching:
discrete observations case, \emph{Quantitative Finance},2013, \textbf{13(5)},
687-698.
- Need to include the limitation of the proposed method and future research work in the conclusion section.
Response 7: We have added some discussion about the limitation of the proposed method and the future work.
However, if the closed form cumulant generating function does not exist, then one needs to rely on a numerical method to find a saddle point. In this situation, the procedure becomes slower.
In the future, it is worth considering multi-state Markov chain case and developing an algorithm to beat the curse of dimensionality. Furthermore, pricing American option under Heston's model with regime switching and its calibration are interesting to study.
- The references are not up to date. Please include the latest 5 years of papers in the references. Please include and discuss the latest research work (year 2018-2021).
Response 8: We have added some up-to-date references.
Round 2
Reviewer 3 Report
The authors have addressed and responded to all the comments given in the review report. Therefore, I have no further comment on this manuscript.